# A BIM-Based Method for Structural Stability Assessment and Emergency Repairs of Large-Panel Buildings Damaged by Military Actions and Explosions: Evidence from Ukraine

Petro Hryhorovskyi [1], Iryna Osadcha [1,2,3,*] 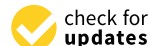, Andrius Jurelionis [3], Vladyslav Basanskyi [1] and Andrii Hryhorovskyi [1]

1   Research Institute of Building Production, 03037 Kyiv, Ukraine
2   Faculty of Construction, Kyiv National University of Construction and Architecture, 03037 Kyiv, Ukraine
3   Faculty of Civil Engineering and Architecture, Kaunas University of Technology, 51365 Kaunas, Lithuania
*   Correspondence: iryna.osadcha@ktu.edu

**Abstract:** The article presents the results of analysis and perspectives for the application of Building Information Modeling (BIM) for the selection of effective organizational, technological, and technical solutions in the elimination of the emergency destruction of large-panel buildings as a result of military actions. It has been established that information–mathematical modeling and the usage of a database on typical destructions can improve the work efficiency and safety of the liquidation of emergency destructions of buildings. Moreover, it enables the scaling and identification of the optimal option of emergency stabilization, as well as organizational, technological, and technical actions that have to be urgently taken to prevent the loss of life or collapse of large-panel buildings after massive damage due to shelling and other military actions. Information–mathematical modeling is explored as the key enabler of assessment and decision-making, while typically there is limited access to the survey object and a lack of information about its condition. The case of a large-panel building in Ukraine damaged as a result of a gas explosion was used for the development of the information–mathematical model and for demonstrating the proposed approach. In post-war times, the use of the presented methodology will allow a rapid assessment of the technical condition of buildings and stabilization strategy selection, including the periodicity of monitoring needs and times of repairs.

**Keywords:** BIM; building destruction; emergency recovery works; information–mathematical modeling; large-panel buildings; progressive collapse risk



## 1. Introduction

Background for the study. Recent developments in Building Information Modeling (BIM), UAV-based surveying, and structural stability simulations enable much faster and more efficient assessment of damage to buildings. As the growing global population and increasing residential housing stock are constantly affected by earthquake damage, slope instabilities, or military actions, it is important to create methodologies that help to tackle these issues by means of novel technologies. Although this issue is globally relevant, such methodologies and solutions should be adopted locally due to specific construction materials, technologies, or types of damage.

This paper focuses on the damage to large-panel residential buildings because of the ongoing war and military actions in Ukraine. According to the Ukraine Rapid Damage and Needs Assessment–August 2022 report that was jointly prepared by the World Bank, the Government of Ukraine, and the European Commission, around 817,000 residential units were impacted by the war [1]. The reconstruction and rebuilding of Ukrainian housing will undoubtedly involve many factors, and novel approaches will be both needed and adopted.

As a result of the military actions in 2022, and at the moment of the publication of this paper, the building stock of Ukraine faces problems related to the elimination of the consequences of the destruction of buildings, which can be categorized into (Figure 1):

(1)     Priority emergency and rescue operations immediately after the rocket-bomb attacks that destroyed the building;

(2)     Planned restoration of objects damaged as a result of military actions employing strengthening, repair, and reconstruction;

(3)     Construction of temporary housing by methods of rapid construction taking the peculiarities of the construction market of Ukraine into account; renovation of existing non-residential buildings and changing their function into residential [2];

(4)     Construction of new housing under post-war rebuilding programs.

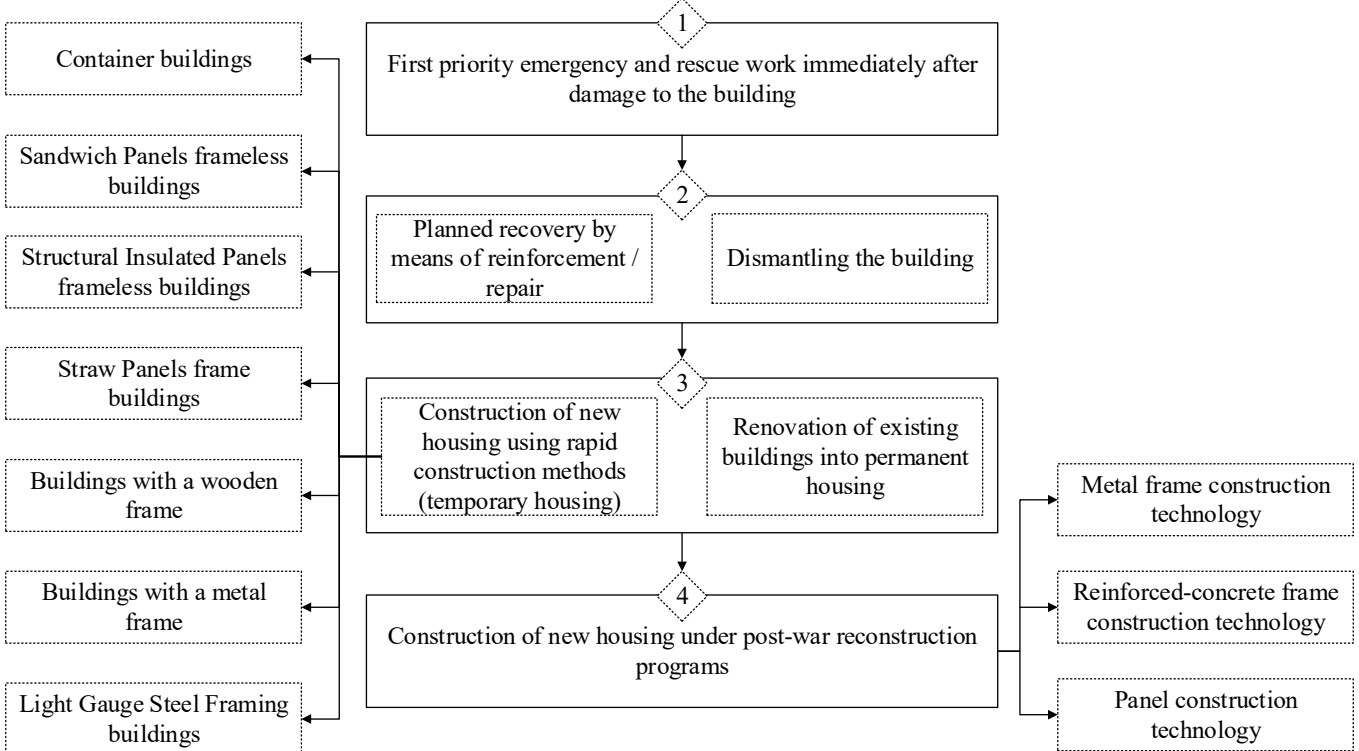

**Figure 1.** The main areas of problems facing the building stock of Ukraine as a result of military actions.

Solving the above-mentioned issues requires a comprehensive study since the number of damaged buildings, at the moment of the submission of this paper, is increasing daily under the conditions of continued hostilities. The situation is in a state of flux, and there is no up-to-date statistical data on the real scale of the destruction in many cities in Ukraine. At this stage, it is expedient to develop mechanisms for the most effective response to new challenges and roadmaps for the post-war rebuilding of cities.

The subject of the research presented in this paper is the first of the indicated areas of construction activity, namely emergency rescue work performed by rescue services under conditions of uncertainty and limited time for decision-making, immediately after the rocket-bomb attacks that damaged the building.

The design of organizational, technological, and technical solutions for the elimination of the consequences of the destruction of buildings due to military actions is a little-studied area [3]. Traditionally, in Ukraine, organizational and technological design, that is, the development of the project of the construction organization and the project of construction works, require the study of engineering research materials, the state of the surrounding environment, the capabilities of the construction organization, its technical base, preparatory work on the construction site, etc., which in turn require time and

preliminary preparation. However, in cases when there is a threat to human life, emergency rescue work is required immediately after a disaster. Therefore, a method to minimize the time and risk of making ineffective decisions under conditions of insufficient information about the damaged object is in high demand.

Among the buildings damaged by military actions, large-panel buildings of mass construction series have a significant share, since buildings of this type form the basis of densely populated residential districts, often bordering industrial zones that suffer from bombings (Figure 2). Therefore, the improvement of organizational, technological, and technical solutions for the elimination of the emergency destruction of large-panel buildings is one of the directions that requires urgent solving.

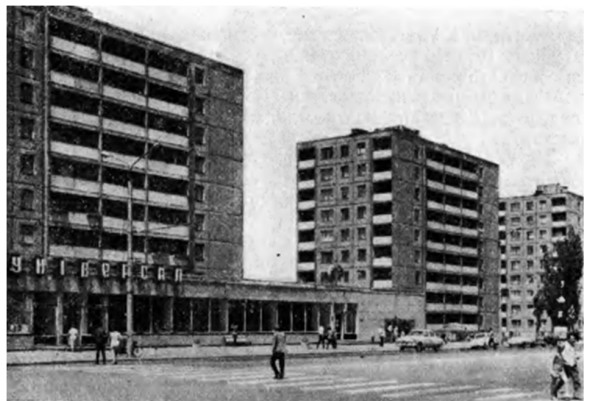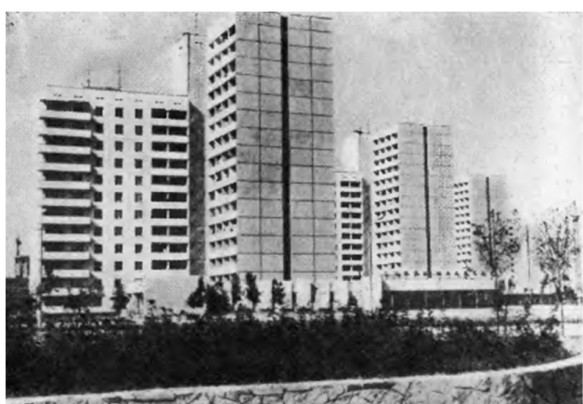

**Figure 2.** Historical images of the development of Dnipro city with large-panel buildings of series 1–164, 1965–1966 [4].

According to the data presented in Table 1, more than 88% of existing residential apartment buildings in Ukraine were built before 1991 [5–8]. About 40% of them are large-panel buildings of mass construction series in industrially developed regions of Ukraine and large cities such as Kyiv, Kharkiv, Dnipro, Zaporizhzhia, etc. (Figure 3). The significant spread of large-panel buildings in Ukraine was one of the reasons for choosing this particular type of building for research.

**Table 1.** The housing stock (apartment buildings) of Ukraine is distributed by years of construction.

| Years of Construction | Share of the Housing Stock within Ukraine | Type and Short Description of Construction | Abbreviations of the Most Common Types (Series) of Panel Buildings |
|---|---|---|---|
| Before 1940 | 6.2% | Historical buildings | – |
| 1941–1960 | 8.9% | Buildings of the early Soviet period ("Stalinky") | – |
| 1961–1970 | 19.6% | Buildings of the first mass series of the period of industrial buildings' construction ("Khrushchovky") | 1–438, 1–464, 1–164–A, 1–480 |
| 1971–1980 1981–1990 | 27.7% 26.1% | Buildings of typical series | I–515/9 m, I–515/9sh, 1605/9, II–18/9, II–29, II–32, II49, 504, BPS, KT, I–134, S–96 |
| 1991–2000 After 2001 | 9.1% 2.4% | Modern buildings | APPS, B–5, ES, KTU |

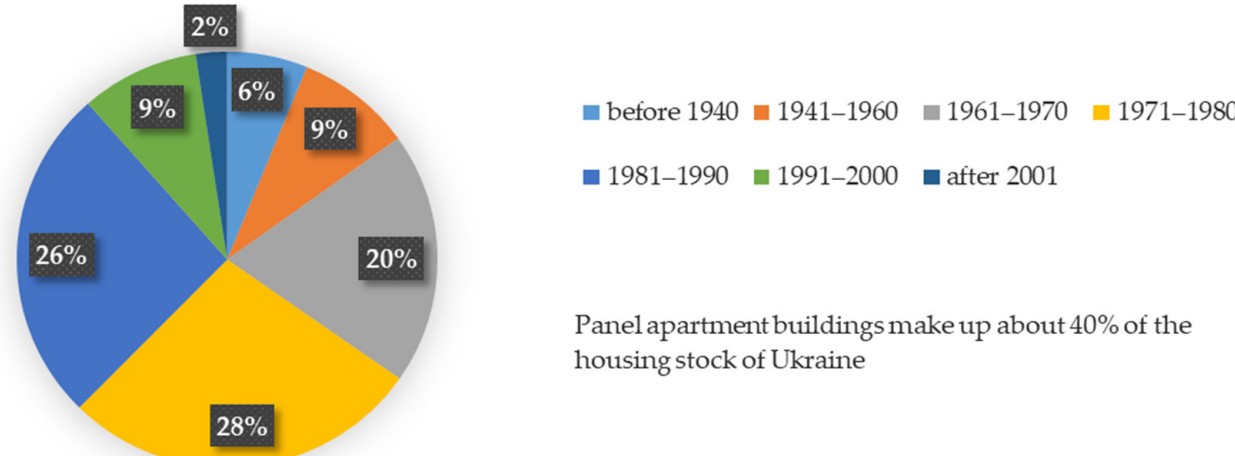

**Figure 3.** The housing stock (apartment buildings) of Ukraine is distributed by years of construction.

The most widespread large-panel buildings in the territory of Ukraine are buildings with the following structural schemes [4]:

- Frameless panels with transverse load-bearing walls: with a small step between the load-bearing walls (2.4–4.2 m), with a large step between the load-bearing walls (4.8–7.2 m), or with a mixed step between the load-bearing walls;
- Frameless panels with longitudinal load-bearing walls: with load-bearing internal and external longitudinal walls or with two load-bearing external longitudinal walls;
- Frame-panel: with a full frame or with a non-full frame;
- Block-panel (modular);
- Panel and frame-panel houses with a monolithic stiffness diaphragm.

Structural schemes of large-panel buildings may differ slightly depending on the series of buildings and years of construction.

As mentioned earlier, around 817,000 residential units were impacted by the war, 38% of them destroyed beyond repair [1]. This number includes apartment units, single family houses, and dormitories. Apartment buildings have been the most affected. The extent of housing damage is spread unevenly across the regions, with the Donetsk, Luhansk, Kharkiv, and Kyiv regions accounting for over 82% of the total damage to the housing stock in the country [1]. Apartment buildings are predominant in urban areas and constitute almost 67% of the urban population. In big cities, this share increases to 79%. Apartment units, particularly Soviet-period apartments, have experienced the bulk of the damage (84%), indicating that the conflict has mainly impacted dense urban areas. [1,9,10].

For example, in Mariupol city, Donetsk region, as of 29 April 2022, 40% of the housing stock was damaged or destroyed [11,12], a significant part of which were large-panel buildings in residential areas (Figure 4). Significant damage and destruction of buildings as a result of hostilities are also common in other bombed cities (Figure 5). As of June 2022, 636 built assets in Kyiv city were damaged [13].

General analysis of the previous technical literature related to the topic. The issue of mass restoration of housing damaged as a result of military actions has been widely studied based on the examples of the post-war reconstruction of European cities after the Second World War, and of developing countries, on the territory where military actions took place recently [14–19]. Using the example of the reconstruction of the city of Mosul in Iraq, after the ISIS war of 2014–2017, the possibility of using BIM for post-war housing reconstruction was considered [20].

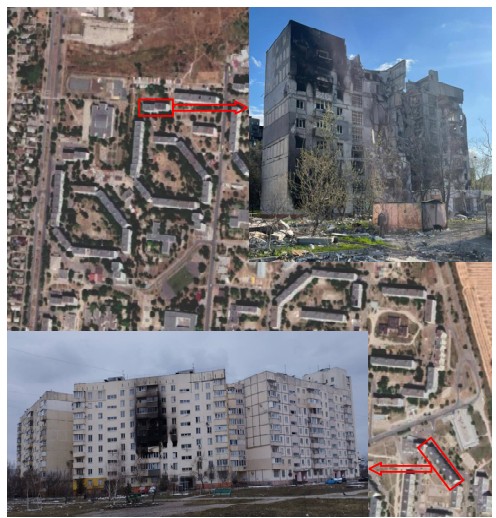 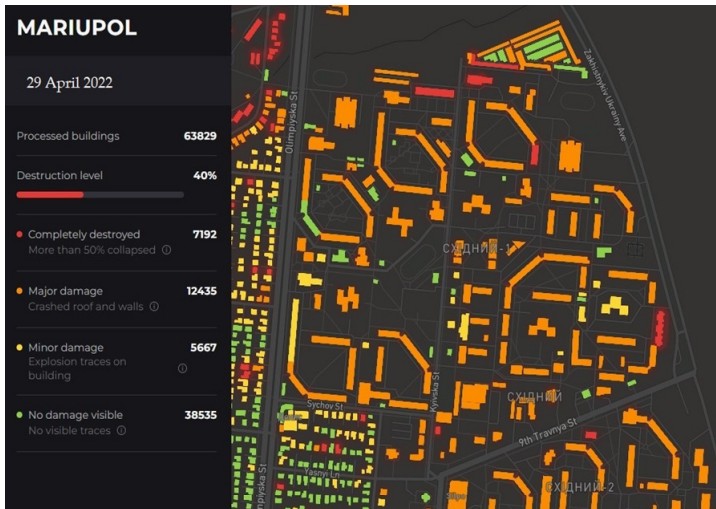

**Figure 4.** Destruction and damage to large-panel buildings of the Skhydnyi district in Mariupol city, Donetsk region [11,12].

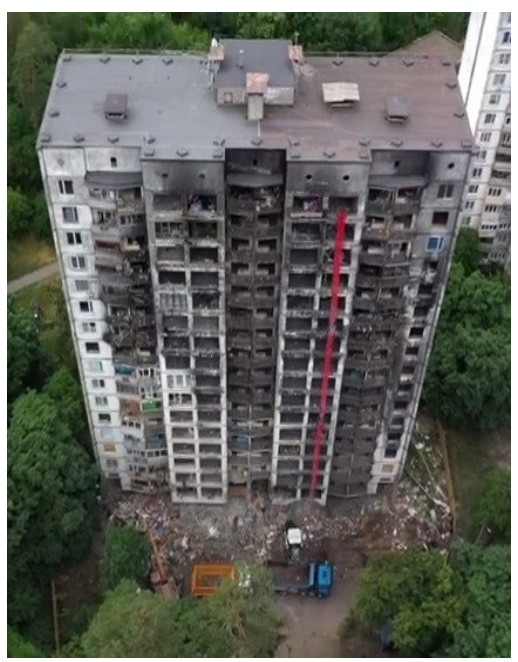 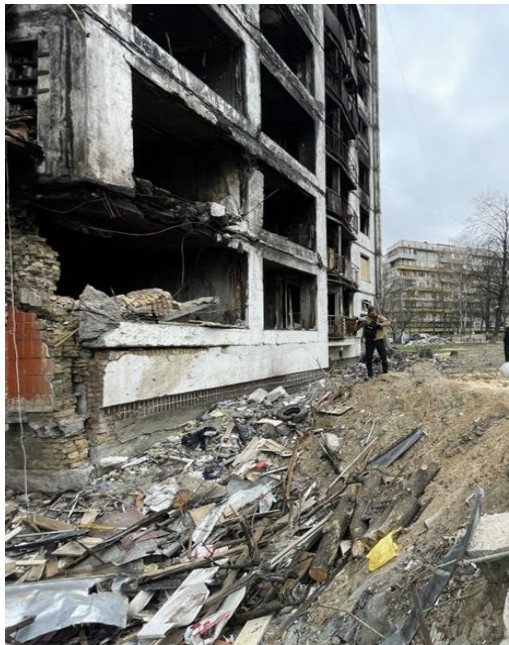

**Figure 5.** Damage to the multi-apartment large-panel building of the BPS series on Chornobylska Street, Kyiv city [13].

The wide implementation of BIM and digital tools in the construction industry allow the finding of new ways of solving urgent problems. In connection with the current study, the possibilities of applying artificial intelligence (AI) and machine learning (ML) to assess the technical condition of buildings after earthquakes are analyzed [21]. The consequences of natural disasters (earthquakes, hurricanes) in terms of the mass and the character of damage to buildings have common features with the consequences of military actions. At the same time, the nature of the damage to buildings is undoubtedly different. A wide range of tools has been developed to assess the state of damage to buildings as a result of natural disasters, which can partially be used to assess buildings damaged as a result of military actions [22–25]. For example, using the Rapid Visual Screening (RVS) method and its modifications [26], Product Lifecycle Data [27] for preliminary assessment of the technical condition of the building, using BIM modeling to evaluate the safety of a

building [28], choosing a building strengthening strategy [29], or assessment of emergency risk for post-damage buildings [30].

The next important Issue is the application of methods and equipment that can provide remote damage assessment. For instance, using TLS (LiDAR) to assess the technical condition of the building and create a 3D model based on the point clouds [31] or UAVs and digital photogrammetry [32,33], as well as automatic damage detection [34], vision-based defect inspection [35], and monitoring the stability of a damaged building using high-resolution laser scanners [36].

Moreover, it is important to assess the stability of the building based on the post-damage survey. This problem was widely covered in publications, especially after the catastrophic events of 11 September 2001 led to the collapse of the WTC towers [37,38], and in order to assess the stability of buildings in seismically active regions [39]. Unfortunately, for prefab buildings, such studies were conducted to a lesser extent, although the study of Munmulla et al., includes an important analysis of the stability of a modular building with various options for removing structural elements [40].

The analysis of previous publications shows that the problem of the urgent inspection of a large number of buildings damaged due to military actions, including large-panel buildings, has not been sufficiently studied until now, and there is still a need for methodologies tackling this issue.

In the context of the mass damage assessment of buildings, most publications focus on the evaluation of damage due to natural disasters. The assessment of the stability of buildings damaged as a result of man-made disasters is not so comprehensive and mainly considers cases of single damage. Accordingly, the methodology of building survey and information–mathematical modeling for decision-making about their stability cannot be fully applied under the conditions of mass damage.

Moreover, the issue of massive man-made damage to residential large-panel buildings was almost not covered in the publications of previous years, as well as the issue of an urgent assessment of the technical condition of such buildings in order to make a decision on the possibility of further use of the building and not evicting residents. Under conditions of massive damage to residential buildings, there is the need to simultaneously relocate a large number of residents and provide them with temporary housing, which is one of the challenges and prerequisites for developing the method of urgent damage assessment.

A similar experience in Ukraine was the accident at the Chornobyl nuclear power plant (ChNPP) in 1986, as a result of which more than 116,000 people from more than 90 settlements were urgently resettled from the "zone of exclusion" to safer regions of Ukraine [41]. In the period of 1986–1987, 23,000 houses, 15,000 apartments, and 800 social and cultural institutions were built for settlers [42]. At the same time, the construction of permanent housing for people affected by the Chornobyl accident in some regions continued until 1995.

Under the conditions of the continuation of hostilities, the number of injured people and forcibly displaced persons urgently in need of housing is much higher. According to the Office of the High Commissioner for Refugees of the United Nations (UNHCR), 11.4 million Ukrainians left their homes only in the first month and a half since the start of the hot phase of the current Russian–Ukrainian war. According to the UN International Organization for Migration (IOM), another 7.7 million citizens of Ukraine are considered internally displaced persons (IDPs); that is, they remained in the country but had to leave their homes [9].

With such a large number of people affected and displaced, the effective use of the existing housing stock is critically important. A quick and most accurate assessment of the operational suitability of damaged buildings, which affects the decision to evacuate or not to evacuate residents, is necessary. Maximizing the use and preservation of the operational suitability of the existing housing stock will reduce the burden and the need for temporary housing and infrastructure for the victims.

Accordingly, the main research question of this investigation is the development of a method for the urgent assessment of the technical condition of large-panel buildings damaged by military actions using information–mathematical modeling in order to reduce the duration of decision-making about their technical suitability. The current paper presents an analysis of the prerequisites for the development of the method, a case study for evaluating the possibilities of information–mathematical modeling, and a workflow for applying information–mathematical modeling to damaged large-panel buildings.

An analysis of publications shows that the topic of the mass inspection of residential large-panel buildings damaged by war actions was never covered before. Existing methods of surveying the technical condition of the building and applying information–mathematical modeling to assess stability can be partially used in the current situation. However, an issue that has not been investigated before is the development of a method for the urgent assessment of the stability of a large-panel residential building, under the conditions of a significant scale of surveys, which will allow for a reduction in the time of surveying, modeling, and decision-making regarding the strengthening and reconstruction of the building or its dismantlement.

Improvements and automatization processes are critical for both the urgent assessment of damage to a large number of buildings and decision-making regarding their stabilization and further operation. A method for such an assessment of large-panel buildings of mass construction series is the focus of this study. The original model for calculating the deformation of the building can be the same for all buildings of the series, with a change in the location and extent of structural damage.

The development of an algorithm for the urgent assessment of the technical condition of a large number of damaged buildings based on a database of typical structural schemes of large-panel buildings, remote assessment of damage, and their subsequent recognition using digital technologies (UAVs (drones), robotics, IoT, cloud-based systems, laser scanning, artificial intelligence, and machine learning) and making changes to the calculation model of the structural scheme will speed up the decision-making process about the possibility and methods of urgent stabilization of the building.

The optimization of the decision-making process under the conditions of uncertainty is possible with the preliminary development of typical organizational and technological emergency measures and the methodology of their use at typical facilities. Linking existing, pre-developed solutions using information–mathematical models of typical objects to a specific emergency object based on the principle of pattern recognition allows the speeding up of the choice of an option and ensures the execution of emergency rescue operations. In turn, this will contribute to the rescue of possible victims, prevent accidents, and become a part of the emergency response plan. The proposed procedure for such assessments is presented in Figure 6.

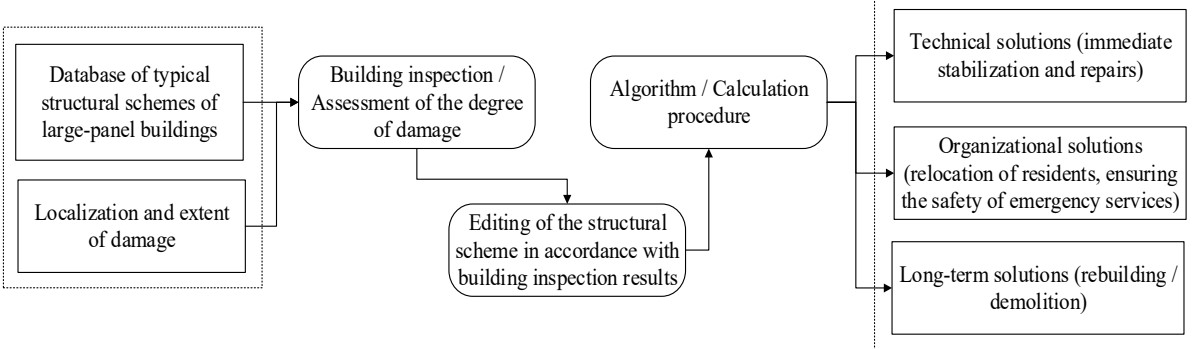

**Figure 6.** Procedure of war damage assessment of and decision-making for the large-panel building.

According to the above, the research objective of this work is to evaluate and analyze the prerequisites for the development of a BIM-based method for structural stability as-

sessments and emergency repairs of large-panel buildings damaged by military actions in Ukraine. The example of a large-panel building damaged by a gas explosion will evaluate the possibility of applying information–mathematical modeling and its influence on decision-making processes about the stability of the building and the need for reinforcement. As a result, a workflow for assessing the technical condition of the damaged building will be developed, along with the definition of its main stages and the possibilities for its improvement or automation.

## 2. Methods and Results of On-Site Measurements and Predictions

At the initial stage of the inspection of a damaged building, it is difficult to obtain objective information about its actual technical condition due to the dangerous and limited access to the object of inspection and the presence of the risk of an unforeseen collapse of the destroyed structures. Accordingly, the use of simulation and information–mathematical modeling [43] can be an effective method of operational probabilistic confirmation of the conclusion about the technical condition of the damaged building in conditions of information uncertainty, when making a decision to prevent the development of the dynamics of destruction is urgent. It enables the making of decisions based on state-of-the-art calculation models and technical characteristics, and can include expert assessment, etc., to obtain predictive data on the stability of structures.

During the primary inspection of the building, immediately after the damage, it should be considered that, at the time of damage to the building, the main factors affecting the structure and leading to destruction are [44]:

- Overpressure in the impact zone;
- Thermal effects that may occur as a result of projectile rupture or fire;
- Debris damage, which can be primary, secondary, or both;
- Energized projectiles that can trigger a further explosion. For most large-panel residential buildings, these are household gas appliances. This was observed after a rocket hit an apartment building on Chornobylska Street, Kyiv city, where a secondary fire broke out (Figure 5). As a result, 64 apartments out of 126 were severely damaged [13];
- Cratering and ground shock, which can provoke the further displacement and collapse of structures.

Therefore, it is important to assess the above listed factors objectively and as soon as possible after the damage has occurred, to prevent further destruction of the building and avoid danger for specialists working at the site.

At the same time, the importance of joints between panels for prefab buildings must be taken into account. For large-panel buildings, joints between panels are one of the most important structural nodes, the use of which is of the greatest importance for the building's operational characteristics and seismic stability.

According to the conditions of the perception of loads and the influence of atmospheric phenomena, it is worth distinguishing two main groups of joints [4]:

- Joints placed in the external walls of the building exposed to atmospheric phenomena, placed between the panels of the external and internal walls and ceilings;
- Joints inside the building, between panels of internal walls and ceilings, and between internal walls.

The joints between the panels of the external walls, in addition to the perception of forces that arise when the structures are compatible and work under the influence of static and dynamic loads, are exposed to the influence of the atmosphere (temperature, humidity, wind). At various stages of the development of large-panel construction in Ukraine, the design of joints has gone through a path of improvement based on the experience of installing structures, operating buildings, and the results of scientific research [4].

In the majority of typical large-panel buildings of the Soviet period in Ukraine, two main types of joints are used: open and closed [4].

In the construction of a closed joint, sealing is carried out by tightly sealing the joint with sealing materials and cement mortar in order to prevent moisture from penetrating into the middle of the panel. For horizontal joints, in addition to laying them with sealing materials, a waterproof ridge is provided in the structure of the panel.

In the design of the open joint, the connection from the outside (except on the first floor) remains open. Barriers to the penetration of moisture into the middle of the vertical joint are drainage strips made of aluminium, propylene, and other materials installed at the depth of the joint. In horizontal plugs, a waterproof ridge is arranged without sealing from the outside.

Depending on the period of construction of the building, different approaches were used to install joints and ensure their sealing [4]. For example, for the series of buildings 1-464 A (1961–1970), which were not built in seismically active regions, the panels were connected by welding metal embedded parts with their subsequent protection from corrosion with a layer of cement mortar. Later, for the buildings of this series, the method of welding the armature releases was used, with the subsequent monolithization of the joints with dense concrete. In the following years, in order to reduce welding work and ensure greater reliability of joints, the method of connecting panels using staples was developed and implemented in mass construction.

Previous work experience (a case study, which will be described below) shows that making an effective decision to ensure the operational suitability of a damaged building requires a detailed analysis of the consequences of damage, the main factors affecting the development of deformation processes, and the risk of progressive collapse, which has an impact on the expected duration and labor costs of implementing measures to stabilize abnormal deformations and ensure the meeting of regulatory requirements for the stability of structural components.

Case study description. The current study, the case of 2020, which occurred in Kyiv city, was chosen as the initial stage of research on the prospects for the application of Building Information Modeling for making operational decisions in the elimination of the emergency destruction of large-panel buildings. The accident in 2020 in Kyiv city due to the gas explosion on the 7th floor of a 10-story large-panel building of the S-96 series showed that there was a need for urgent stabilization of the building to prevent further destruction.

Based on this case, the effectiveness of information–mathematical modeling at the stage of emergency work immediately after building damage is analyzed. Data on structural deformations is essential to predict further displacement of structural elements and to develop technical solutions for building stabilization. The workflow that was applied for assessing the technical condition of a large-panel building after a gas explosion and making a decision is shown in Figure 7.

On the basis of the results of an urgent inspection of the building, available materials about the structural scheme, such as floor plans (Figure 8a), and information about the technical condition of the building during its operation stage, information–mathematical modeling and stress–strain state analysis were performed. To analyze the stress–strain state of the building, the calculation scheme Finite Element Method (FEM) of the building with the removal of collapsed structural elements was created in the software complex MONOMAH CAD (Figure 8b).

Characteristic structure weight loads, long-term loads, short-term loads, and horizontal operational wind loads were in the design scheme of the building. An example of vertical loads on floor slabs of the first floor is shown in Figure 8a. The weight of the structure was taken into account automatically. Wind loads were specified and complied with the regulatory documents in force in Ukraine and were as follows: wind area = 2, pressure $W_o = 0.045$ ts/m$^2$, type of terrain = II, coefficient of geographical height Calt = 1.4, coefficient of dynamism $C_d = 1.2$, and coefficient of reliability according to operational value $Y_{fe} = 0.21$.

Since the technical condition of structural elements and joints is critically important for a full assessment of the magnitude of forces and deformations of the building frame, this

was taken into account in the calculation model. At the same time, in general cases, either a reduction in the cross-section of the joint points or a reduction in the strength indicators can be applied in accordance with the values determined by the results of the technical inspection or predicted values.

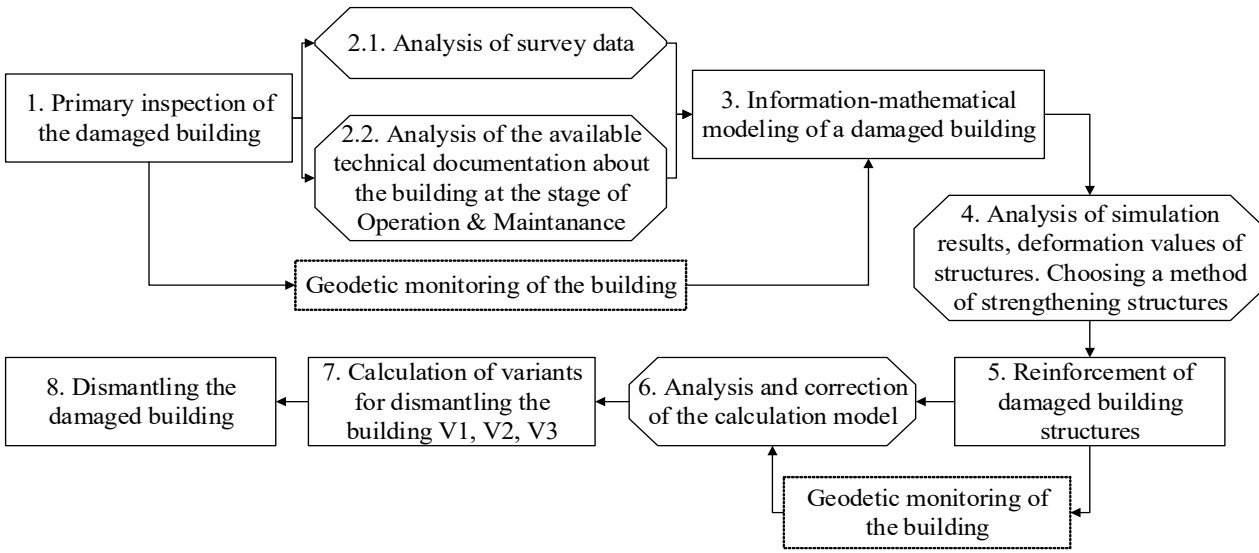

**Figure 7.** Workflow of the large-panel building damage assessment and decision-making for the case study.

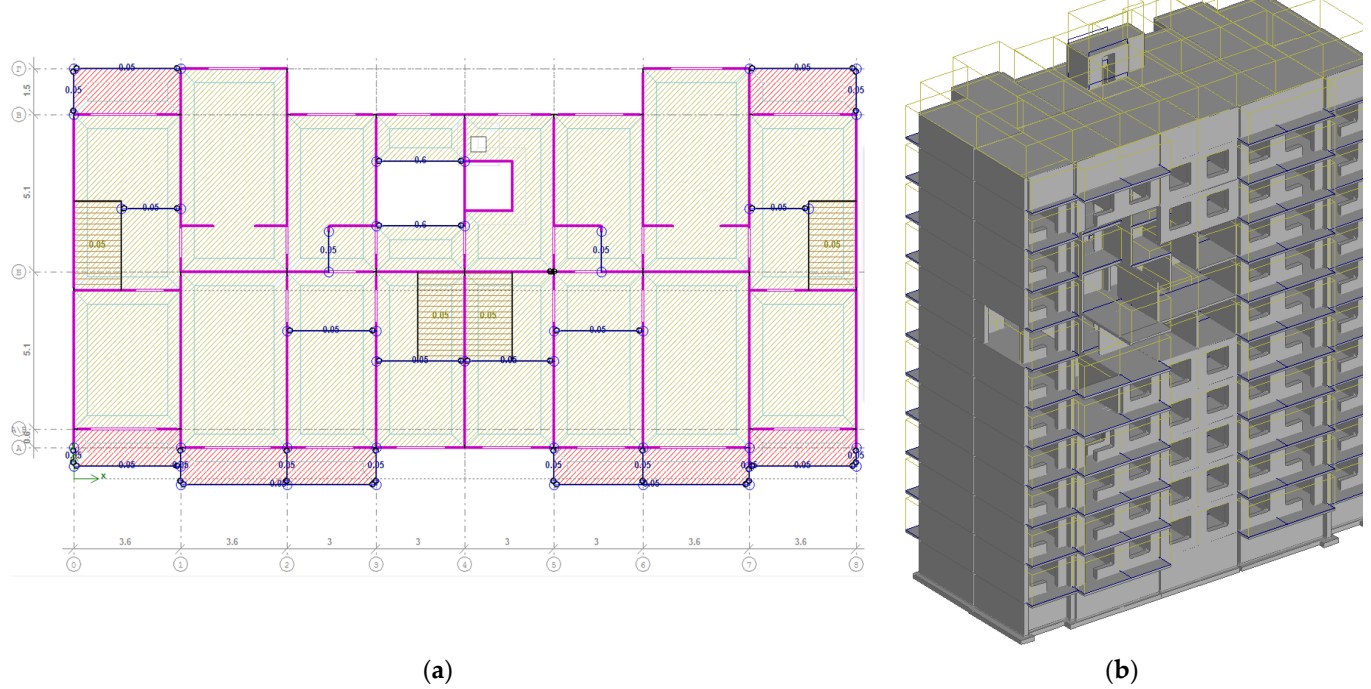

(**a**)                                        (**b**)

**Figure 8.** Creation of the spatial calculation scheme of the building: (**a**) plan of the first floor with given vertical loads of the floor slab; (**b**) 3D view of the structural scheme of the building with removed damaged elements (Kyiv city, S. Krushelnytska Street, June–October 2020).

A preliminary analysis of the stress–strain state of the building was performed on the structure weight and long-term loads after the removal of the collapsed elements of the structures. The isofield of stresses and displacement along the X, Y, and Z axes was

determined (Figure 9a). The results of the calculations and the obtained deformation values proved that the further operation of the building section was impossible and that it should be dismantled.

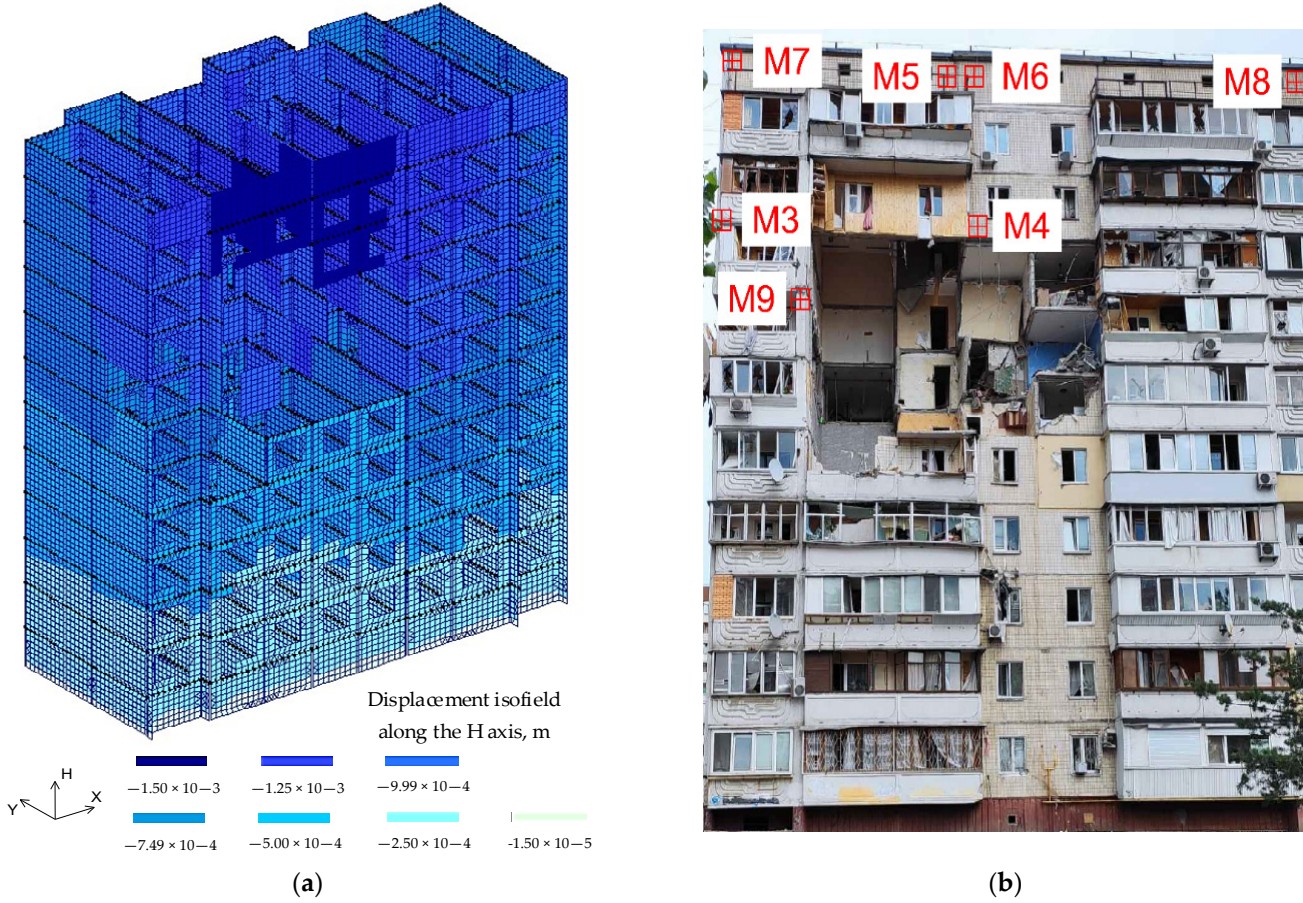

**Figure 9.** The results of frame deformation predictions of a large-panel building (**a**) and the scheme of the placement of observation points to the deformations of the structures (**b**) (Kyiv city, S.Krushelnytska Street, June–October 2020).

At the next stage, the FEM calculation scheme was exported to the software complex LIRA CAD to consider the modeling of the stages of dismantling the variable structural system of the building. According to the results, the necessity of adding reinforcement elements (frame elements) to avoid collapse, which may affect adjacent sections, was determined. The installation of these elements stabilized the horizontal and vertical deformations of the building, but the building remained in an emergency state. Considering that the reinforcement structures were installed for a short time during the dismantlement of the building, and their permanent operation was not expected, their use as temporary supports could be allowed (Figure 10). Such an approach enabled the performance of emergency work and a further detailed inspection of the building.

After performing strengthening elements, the spatial stability of the building increased, but considering the resulting total deformation from the explosion, the building continued to be in a state of unstable equilibrium. Therefore, before the start of the dismantlement and from the beginning of work, it was necessary to carry out operational constant control of the deformations and distortions of the building.

Moreover, periodic geodetic monitoring was carried out to control the stability of the building and ensure the reconstruction work safety. Measurement of structural deformations and assessment of their dynamics were ensured by observing the deformation marks (M1–M9) placed on the facade of the building (Figure 9b). The first set of observa-

tions was carried out during the first days after the explosion and damage to the building (June–July 2020), and the next after strengthening the structures and ensuring the temporary stability of the building (September–October 2020). The complete data of geodetic observations of deformation marks with the indicated dates and times of observations are shown in Appendix A.

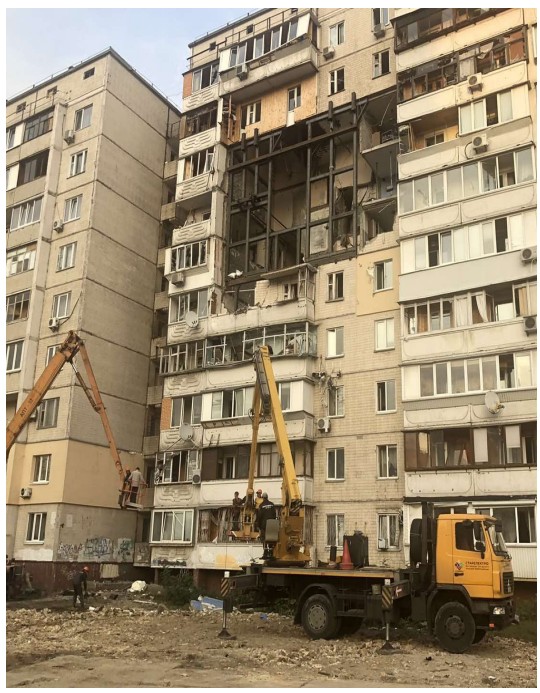 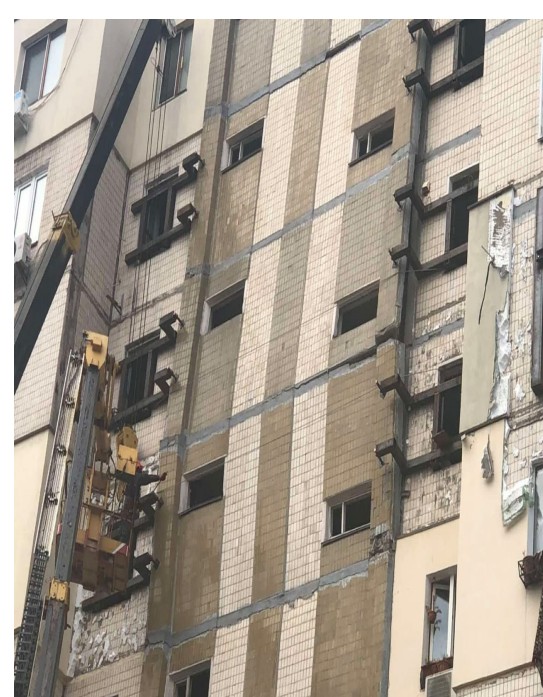

**Figure 10.** Urgent temporary stabilization of a large-panel building damaged as a result of a gas explosion (Kyiv city, S. Krushelnytska Street, June–October 2020).

The analysis of the obtained data (Figure 11) shows the deformation dynamics for each of the axes (the orientation of the coordinate system is shown in Figure 9a). The value of the deviation of deformation marks for each cycle of observations was determined as the difference between the coordinates measured in the current cycle and the coordinates of the deformation mark at the time of its installation (0-cycle):

$$\Delta_{n(X,Y,H)} = C_{n(X,Y,H)} - C_{0(X,Y,H)}, \tag{1}$$

where: $\Delta_{n(X,Y,H)}$—the value of the change in the coordinates of the deformation mark in the current cycle of observations; $C_{n(X,Y,H)}$—coordinates of the deformation mark in the current cycle of observations; and $C_{0(X,Y,H)}$—initial coordinates of the deformation mark.

The determination of the coordinates of the deformation marks is influenced by a combination of factors: weather conditions, the reliability of fixing the marks, the method of performing work and the equipment used, deformation of the structural elements on which the marks are installed due to damage, displacement of structural elements on which deformation marks are installed as a result of strengthening works, etc. In the current case study, the main purpose of geodetic observations was to monitor the dynamics of development and control deformations in order to prevent the uncontrolled collapse of structures at the time of work. Geodetic monitoring, as an additional method of controlling the stability of the building, confirmed that the chosen method of strengthening the structures provided the necessary time and safe conditions for a detailed survey of the damaged building by specialists and increased the safety of residents during evacuation.



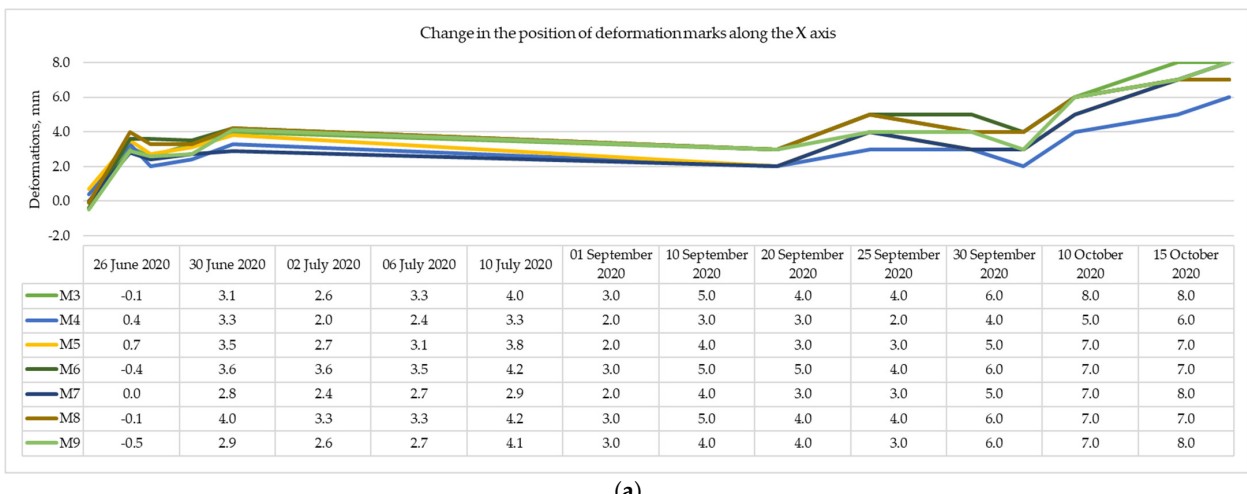

(a)

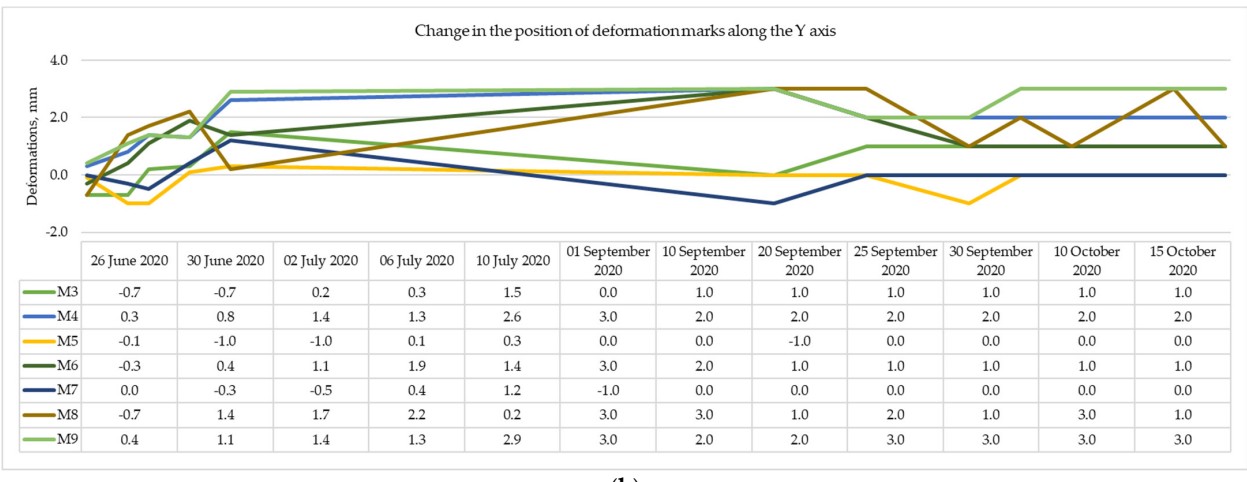

(b)

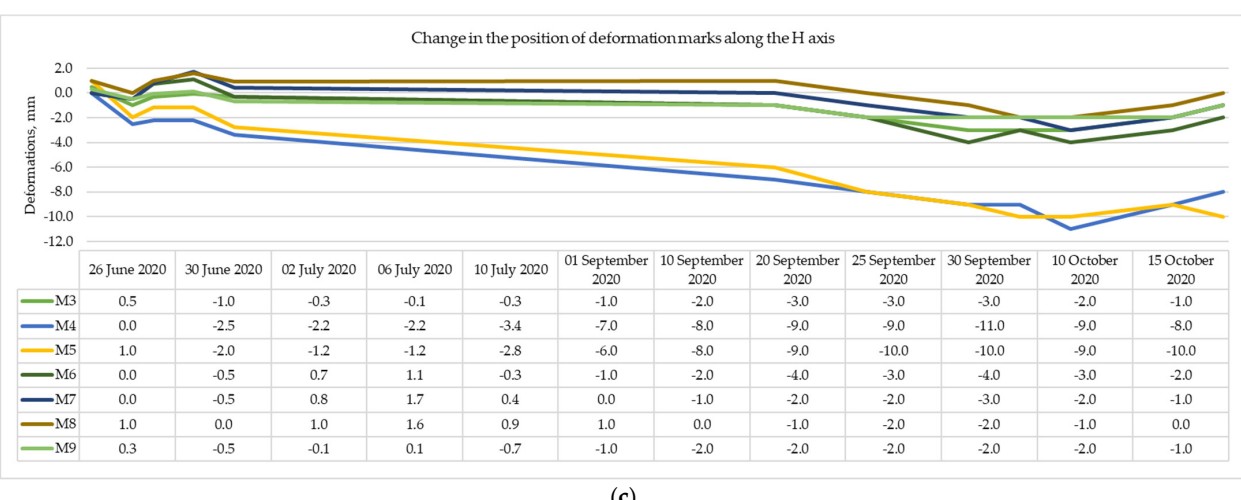

(c)

**Figure 11.** Graphs of subsidence of deformation marks fixed on the facade of a large-panel building damaged as a result of a gas explosion (Kyiv city, S. Krushelnytska Street). Two sets of observations: immediately after the explosion (June–July 2020) and after strengthening the structures (September–October 2020). Change in position of deformation marks along the X axis (**a**), Y axis (**b**), H axis (**c**).

The informational–mathematical model was refined after a detailed survey of the damaged building. An analysis of the defects and damage that changed the main design and boundary conditions for the calculation was conducted. This included characteristics

of structures, determination of actual operating loads and effects on building structures, displacements in bases and foundations, and building services systems. Detailed geodetic measurement of the necessary geometric parameters of the building was carried out, and verification calculations of the main load-bearing building structures, engineering systems, foundation, and object as a whole were performed. The analysis of the detailed data on the defects and damage enabled a more precise estimation of the technical conditions of building structures and engineering systems. As a result, the section of the building was deemed unfit for further operation and was dismantled [45].

The result of the analysis of the case study. The above case study shows that information–mathematical modeling combined with displacement and deformation monitoring can be used for both the initial assessment and continuous monitoring of the building structural stability for residential large-panel buildings. In the current situation in Ukraine, most decisions on emergency stabilization and determination of the further operational suitability of the building are made under the pressure of time and with insufficient information. However, having typical large-panel buildings models would help to increase the precision of structural behavior and reduce surveying and modeling time.

Due to the lack of time to compile an information–mathematical and calculation model, and considering the level of devastation, it is expedient to develop a database of typical BIM-based building models and accumulate information on typical damages, as well as typical organizational, technological, and technical solutions to speed-up the decision-making process. The timeliness and objectivity of the selection of the stabilization actions using information–mathematical modeling are required to avoid the progress of the building destruction. However, due to the large number of damaged buildings, the method used in the case study needs to be supplemented, especially in terms of building survey.

Inspection of the building after damage. In the case of limited access to the damaged building, it is advisable to supplement the existing types of inspections with another type, i.e., a preliminary remote assessment of the technical conditions of structures. A preliminary remote assessment of the technical condition of the building must be carried out in the event of a threat of unforeseeable collapse of structures and, as a result, limited access to the inspection object.

Remote observations can be carried out with the help of laser scanners, UAVs (drones), robotic total stations, etc. A preliminary assessment of the damaged building is performed based on the available raw data obtained remotely: photo and video recording materials, existing technical documentation, floor plans, facades, and sectional drawings.

According to the results of remote surveys for a large-panel building of a typical series, it is possible to develop preliminary (tentative) BIM-based calculation models for the formation of preliminary recommendations regarding the danger of progressive destruction considering the received damage. Unfortunately, at present, such decisions are made intuitively, based on the experience of specialists performing emergency and rescue work. Their actions are a perceived risk that can lead to unforeseen consequences.

At the next stages of the inspection of the emergency building, which include visual inspection and detailed instrumental inspection, the information model has to be refined [45,46]. Only in the case of indisputable certainty about the critical nature of the damage and the predicted progressive collapse can the building be considered beyond repair. Provided this is not confirmed by predictions, or if there is still a lack of information to make a reasoned decision, an additional visual and detailed instrumental inspection should be performed.

Based on the detailed inspection, locations of defects should be noted, the information–mathematical calculation model should be refined, and simulations have to be rerun. After strengthening and repair actions are defined, an estimate should be drawn up and the cost of restoration work should be estimated.

In the case of more complex destructions, initial data are obtained for designing the restoration of the building through major repairs or reconstruction. To do this, an instrumental survey is needed with the establishment of the physical and technical characteristics of the structural components of the building and its joints. An instrumental survey (geode-

tic observation of building deformations, assessment of the condition of structures and joints by non-destructive methods, etc.) should be carried out in volumes sufficient for designing measures to restore operational suitability, considering the results of the previous stages of the survey and preliminary calculations.

Calculations of the bearing capacity of the building as a whole and the development of solutions for strengthening and repair should be carried out using an information–mathematical model [47–49], refined on the basis of the results of instrumental surveys and dimensional drawings, taking the established physical and technical characteristics of the building structures, based on the results of a detailed survey, into account, which involves obtaining all the necessary information for the development of the building restoration project (Figure 12).

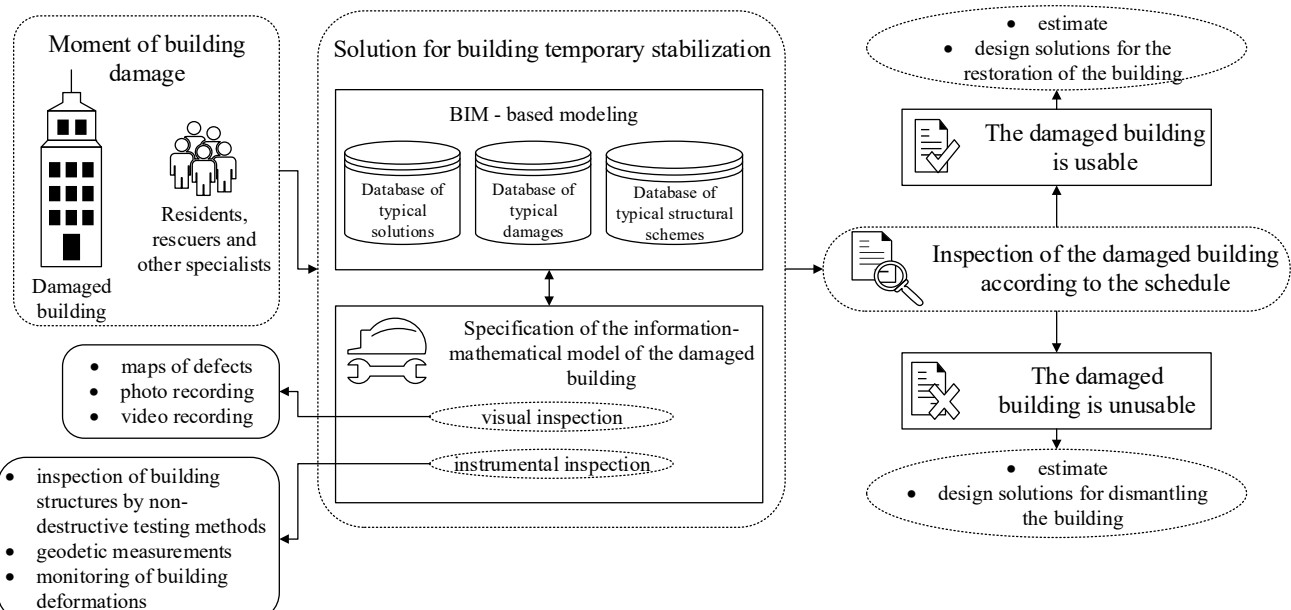

**Figure 12.** The proposed conceptual scheme for assessing the technical condition of a damaged large-panel building using a database of typical solutions and a preliminary remote assessment at the stage of emergency and rescue operations.

## 3. A Method for BIM-Based Structural Assessment

The analysis and study of measures to strengthen and ensure the bearing capacity of the emergency structural components of a large-panel building damaged by gas explosion in 2020 showed that an effective BIM-based workflow is not only needed, but it can also provide the required level of information for decision-making in war-affected areas.

At this stage, we have analyzed the organizational and technical methods that are mainly used to eliminate the consequences of emergencies and strengthen emergency structural components [50]. It has been established that their systematization and timely updating will allow the development of an information base of the typical structural schemes of large-panel buildings in the region, damage options, and methods of urgent stabilization, which can be used for information–mathematical modeling of damaged buildings at the stage of emergency and rescue operations to prevent the development of abnormal deformations.

To increase the efficiency and safety of works, as well as to reduce the risk of progressive building collapse, the improvement of organizational, technological, and technical solutions for the elimination of the damage is needed. This should lead to an effective option for stabilization and preventing the development of excessive deformations, as well as a calculation engine for modeling the synergistic effect of the multifactorial impact of a set of abnormal and man-made factors on the strength and stability of the building, taking

changes in its structural scheme, depending on the extent and localization of damage, into account. The proposed BIM-based workflow for the assessment of damaged large-panel buildings with identified processes that potentially could be automated is shown in Figure 13.

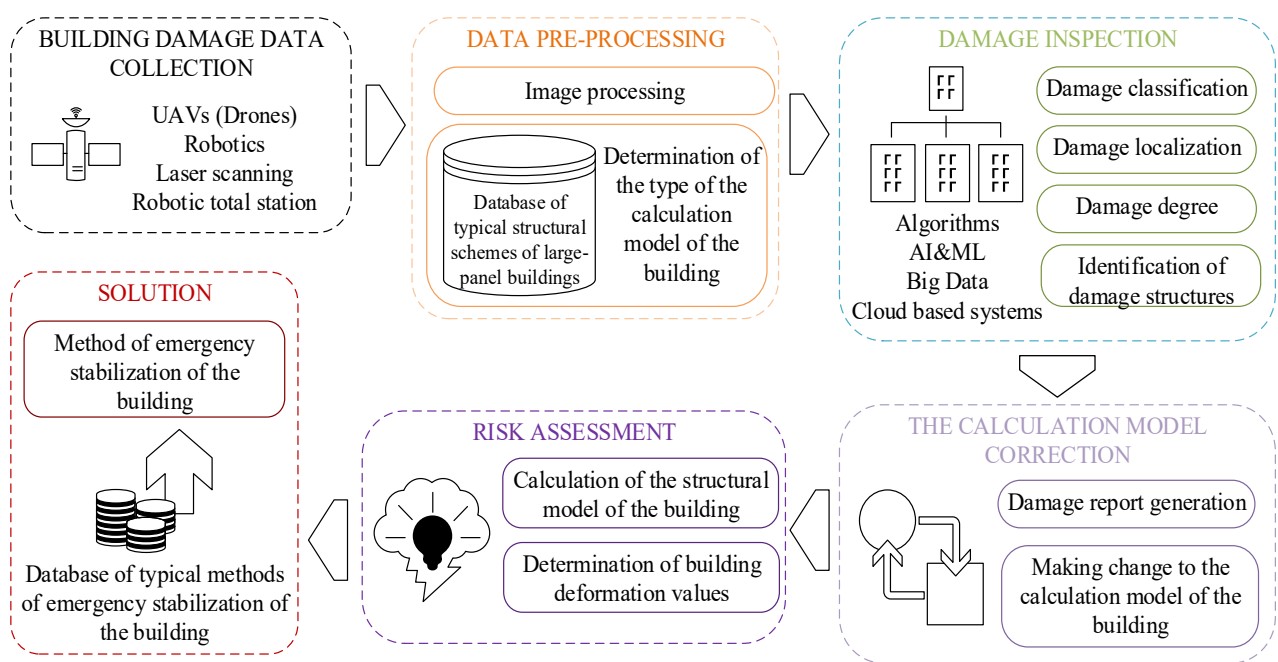

**Figure 13.** Proposed BIM-based workflow for the assessment of damaged large-panel buildings with identified processes that potentially could be automated.

Accordingly, the Building damage data collection block is described in detail in the Inspection of the building after damage section of this paper.

Data pre-processing includes the pre-processing of remote survey data and existing building materials to determine building type and structural scheme.

Damage inspection assesses localization, degree of damage, and damaged structural elements. Therefore, at this stage, there may be a high need for the automation of the process of determining damage based on digital technologies.

The calculation model correction includes creation of an information–mathematical model, and the extraction of damaged structural elements to assess the stability of the building. For this purpose, the following methods can be used to build an information–mathematical model: the Finite Element Method (FEM), Discrete Element Method (DEM), Applied Element Method (AEM), or Cohesive Element Method (CEM). The application of the Finite Element Method (FEM) is the most appropriate due to its wide application in assessing the possibility of the progressive collapse of the building [51].

Risk assessment includes the calculation of deformations and the possibility of the development of an uncontrolled progressive collapse of the building. Progressive collapse is a collapse that begins with localized damage to one or more structural components and develops throughout the structural system, affecting other components [51–53].

In the general case, the assessment of the progressive collapse of the building is performed according to the equation:

$$P(C) = P(C \mid DH) \times P(D \mid H) \times P(H), \qquad (2)$$

where: $P(C)$–the probability of progressive collapse; $P(C \mid DH)$–the probability of progressive collapse C of the structure as a result of local damage D caused by hazard H; $P(D \mid H)$–the probability of local damage D as a result of a hazard H; and $P(H)$–the probability of the occurrence of a hazard H [53–55].

Accordingly, the probability of progressive collapse can be minimized in three ways: controlling abnormal events, controlling local element behavior, and/or controlling global system behavior [56].

When assessing the risks, it is necessary to consider that the progressive collapse of the building can develop according to one of the types or their combination. Starossek (2007) defined six types of progressive collapse: pancake-, zipper-, domino-, section-, instability-, and mixed-type collapses [57].

When assessing the risks of an uncontrolled progressive collapse of a building, a detailed analysis of the deformation values obtained as a result of the model calculation is important. However, not all factors affecting the development of deformations can be fully assessed due to a lack of time and limited access to the building. For example, the quality of panel joints and their condition at the time of building damage, which also depends on the duration of operation, affect the overall stability of the building and its structures [58–61]. The determination of the current technical condition of joints can be performed at the stage of a detailed inspection of the building using non-destructive testing methods. It is especially important to determine the state of joints in places close to the epicenter of damage in order to prevent the uncontrolled collapse of structures. At the same time, in some cases, performing these works can be dangerous and possible only after the stabilization and temporary strengthening of the building.

Considering all the above, cases for which the calculated values of deformations are close to or exceed the normative values should be considered in more detail and carefully.

The Solution block includes possible options for the stabilization and strengthening of building structures based on preliminary damage analysis and assessment of the risk of progressive collapse. For residential large-panel buildings, typical solutions can be developed based on the analysis of the most common structural schemes.

As a result of predictions obtained from the information–mathematical model, urgent decisions to minimize risks in conditions of uncertainty, have to be taken to select effective variants of solutions to address an emergency. The effectiveness of the adopted decision involves the optimization of labor and financial costs, provided that the normative requirements for the stability of the building are met. In the future, the information–mathematical model that was created to assess the stability of the building can be used as the basis for the development of a digital twin, which can be one of the directions of further development of the method. Accordingly, the proposed workflow expands the capabilities of the large-panel building damage assessment method considered in the case study. It can potentially increase its effectiveness in the case of an urgent survey of a large number of damaged buildings.

Further development of the conceptual scheme of the method proposed in this paper, with the definition of detailed requirements for the description of data and the structure of the database (typical structural schemes of large-panel buildings and typical solutions for their stabilization and structural strengthening), as well as their development, are the subject of further research in this direction.

## 4. Discussion

At the time of the submission of this paper, the war in Ukraine is still undergoing. Therefore, there are limitations both in terms of data acquisition and the validation of the proposed workflow. Issues that need to be resolved in the creation of an information database of typical structural schemes of large-panel buildings in the region, including options for building damage and typical organizational, technological, and technical solutions for urgent stabilization of the building, are as follow:

- A lack of up-to-date statistics on the actual number of damaged buildings in each of the regions of Ukraine;
- The introduction of military actions, a change in the dynamics of the situation, which prevents a full assessment;

- Information on the structural schemes of large-panel buildings and their technical conditions before the start of hostilities is fragmented, incomplete, and, in most cases, on paper;
- The lack of a sufficient number of qualified personnel who can be involved in the inspection of damaged buildings in the region, as a result of forced relocation to safer regions of Ukraine or abroad. The complicated work of state authorities and state institutions as a result of martial law in Ukraine;
- The determination of technical requirements for the information database, and criteria for assessing the quality and duration of works.

Unfortunately, some of these restrictions can be fully resolved only after the stabilization of the situation in the post-war period. Nevertheless, primary theoretical works and their experimental implementation in territories where there are no active hostilities should be carried out and will be considered in further research.

## 5. Conclusions

1. The prerequisites for the development of the BIM-based method for the structural stability assessment and emergency repairs of large-panel buildings damaged by military actions were analyzed. It was established that existing methods of surveying damaged buildings can only partially be employed to solve the current problem. Accordingly, the development of a method that will allow for a reduction in the time of surveying, modeling, and decision-making regarding the reconstruction or dismantlement of the building, under the conditions of massive damage to residential large-panel buildings in Ukraine, is an issue that needs to be resolved.
2. Using the example of a large-panel building damaged as a result of a gas explosion, the possibility of using information–mathematical modeling to assess the technical condition and make a decision about the stability of the building was considered. The main stages of the method applied in the case study were determined.
3. It was proposed to supplement the stage of the urgent inspection of the building after the explosion with remote inspections in order to reduce the duration and increase the safety of works. It was determined that, under the conditions of mass damage to buildings, it is expedient to develop a database of typical structural schemes of large-panel buildings, which will reduce the time of simulation.
4. A novel BIM-based workflow for the assessment of damaged large-panel buildings with identified processes that could potentially be automated was proposed. The key issues requiring further research have been identified, the first of which is the development of requirements for data description and database structure (database of typical structural schemes of large-panel buildings and typical solutions for their stabilization and structural strengthening). In post-war times, the development of BIM models could feed into the methodology proposed in this paper. Strengthening solutions can be standardized and constantly updated models could be used for monitoring their technical conditions. This will be considered in further research.

**Author Contributions:** All signing authors participated in all of the writing processes of the article. All authors have read and agreed to the published version of the manuscript.

**Funding:** This research received no external funding.

**Institutional Review Board Statement:** Not applicable.

**Informed Consent Statement:** Not applicable.

**Data Availability Statement:** Not applicable.

**Acknowledgments:** The authors of this paper acknowledge the support of the Marius Jakulis Jason Foundation and the Lithuanian Scientific Society in this research and express their gratitude for providing the opportunity to continue research at the Kaunas University of Technology.

**Conflicts of Interest:** The authors declare no conflict of interest.

## Appendix A

The table of the movements of deformation marks, fixed on the facade of the building, along each of the coordinate axes. After the explosion, the building was monitored twice a day to determine the influence of ambient temperature on the dynamics of the movement of deformation marks.

| Point name | | M3 | | | M4 | | | M5 | | | M6 | | | M7 | | | M8 | | | M9 | | |
|---|---|---|---|---|---|---|---|---|---|---|---|---|---|---|---|---|---|---|---|---|---|---|
| Date | Time | ΔX | ΔY | ΔH | ΔX | ΔY | ΔH | ΔX | ΔY | ΔH | ΔX | ΔY | ΔH | ΔX | ΔY | ΔH | ΔX | ΔY | ΔH | ΔX | ΔY | ΔH |
| 26 June 2020 | 13:00 | -0.2 | -0.1 | 0.2 | -0.3 | 0.0 | 0.5 | 0.0 | 0.0 | 0.0 | 0.3 | 0.0 | -0.3 | – | – | – | – | – | – | – | – | – |
|  | 15:00 | -0.5 | -0.4 | 0.0 | -0.2 | -0.2 | 0.3 | -0.3 | -0.7 | 0.3 | 0.0 | 0.0 | 0.0 | -0.2 | -0.2 | 0.3 | -0.2 | -0.2 | 0.0 | – | – | – |
|  | 16:30 | -0.1 | -0.7 | 0.5 | 0.4 | 0.3 | 0.0 | 0.7 | -0.1 | 1.0 | -0.4 | -0.3 | 0.0 | 0.0 | 0.0 | 0.0 | -0.1 | -0.7 | 1.0 | -0.5 | 0.4 | 0.3 |
| 27 June 2020 | 8:30 | 0.2 | -1.8 | -0.5 | -0.1 | -0.7 | -2.0 | 0.5 | -2.2 | -1.0 | 0.3 | -1.1 | -0.5 | -0.9 | -2.0 | -0.5 | -0.1 | -0.7 | 1.0 | 0.4 | 0.3 | 0.0 |
|  | 17:45 | 0.8 | -1.0 | 1.3 | 0.4 | -0.5 | -0.7 | 1.1 | -1.3 | 0.0 | 1.0 | 0.4 | 1.0 | 0.6 | -2.8 | 1.3 | 0.4 | -0.5 | 0.7 | -0.5 | -0.4 | 0.7 |
| 28 June 2020 | 8:00 | 0.2 | -0.7 | 0.7 | 0.0 | 0.0 | -0.3 | 0.1 | -1.6 | -0.3 | 0.8 | -1.0 | 0.7 | -0.6 | -1.4 | 0.7 | 0.7 | -0.3 | 0.7 | -0.2 | -0.2 | 0.3 |
|  | 15:30 | -0.5 | -1.0 | 0.3 | 0.0 | -0.4 | -2.0 | 0.4 | -2.0 | -0.5 | 0.1 | -0.9 | 0.8 | -0.9 | -1.8 | 1.3 | -0.3 | 0.4 | 0.5 | -0.4 | 0.0 | 0.3 |
| 29 June 2020 | 7:30 | 0.6 | -0.8 | -2.0 | 1.4 | -0.1 | -4.0 | 0.9 | -1.9 | -3.0 | 1.4 | -0.1 | -2.0 | 0.0 | 0.0 | 1.5 | 1.4 | -0.1 | -1.0 | 0.4 | 0.3 | -1.5 |
|  | 17:30 | -0.1 | -0.7 | -0.5 | -0.4 | -0.3 | -2.0 | 0.8 | -2.6 | -1.5 | 0.6 | -1.5 | 1.5 | -1.8 | -4.1 | -0.5 | 0.6 | -1.5 | 0.0 | -1.2 | -1.0 | -0.5 |
| 30 June 2020 | 7:00 | 3.1 | -0.7 | -1.0 | 3.3 | 0.8 | -2.5 | 3.5 | -1.0 | -2.0 | 3.6 | 0.4 | -0.5 | 2.8 | -0.3 | -0.5 | 4.0 | 1.4 | 0.0 | 2.9 | 1.1 | -0.5 |
|  | 19:00 | 0.8 | -3.3 | 0.5 | 3.9 | 0.0 | -3.0 | 3.8 | -1.4 | -1.5 | 4.6 | -0.8 | -0.5 | 3.5 | -1.0 | -0.5 | 4.0 | 0.7 | 0.5 | 4.0 | 0.7 | -1.0 |
| 01 July 2020 | 8:00 | 1.7 | -1.2 | -0.5 | 2.0 | -0.9 | -2.5 | 1.5 | -2.6 | 1.5 | 2.1 | -0.2 | -1.5 | 1.0 | -1.2 | 0.5 | 1.4 | -0.1 | 1.0 | 1.5 | 0.6 | 0.0 |
| 02 July 2020 | 7:35 | 2.6 | 0.2 | -0.3 | 2.0 | 1.4 | -2.2 | 2.7 | -1.0 | -1.2 | 3.6 | 1.1 | 0.7 | 2.4 | -0.5 | 0.8 | 3.3 | 1.7 | 1.0 | 2.6 | 1.4 | -0.1 |
| 03 July 2020 | 6:42 | 3.0 | 0.7 | 0.1 | 2.7 | 2.0 | -1.8 | 2.7 | -0.3 | -0.1 | 3.0 | 1.8 | 1.5 | 2.5 | -0.6 | 1.0 | 2.8 | 2.7 | 1.3 | 2.2 | 2.0 | 0.3 |
| 04 July 2020 | 6:57 | 3.0 | -0.2 | 0.3 | 2.7 | 1.6 | -2.3 | 3.0 | -0.6 | -0.8 | 3.7 | 1.1 | 0.5 | 2.2 | -0.3 | 1.2 | 3.3 | 2.6 | 1.6 | 3.1 | 1.7 | 0.4 |
| 05 July 2020 | 6:49 | 3.1 | -0.4 | 0.5 | 3.1 | 0.8 | -2.1 | 3.5 | -0.7 | -0.4 | 4.1 | 1.9 | 0.5 | 2.6 | 0.0 | 2.2 | 4.2 | 2.1 | 1.7 | 3.1 | 1.7 | 0.6 |
| 06 July 2020 | 7:09 | 3.3 | 0.3 | -0.1 | 2.4 | 1.3 | -2.2 | 3.1 | 0.1 | -1.2 | 3.5 | 1.9 | 1.1 | 2.7 | 0.4 | 1.7 | 3.3 | 2.2 | 1.6 | 2.7 | 1.3 | 0.1 |
| 07 July 2020 | 6:55 | 3.1 | -0.5 | 1.4 | 2.5 | 0.6 | -1.3 | 3.6 | -0.9 | -0.6 | 3.7 | 0.7 | 2.7 | 2.3 | -0.9 | 2.3 | 4.2 | 1.4 | 2.3 | 2.9 | 0.6 | 1.0 |
| 08 July 2020 | 12:30 | 1.4 | 1.6 | -1.2 | 1.1 | 2.0 | -4.6 | 0.8 | 0.1 | -4.3 | 1.3 | 0.9 | -2.0 | 0.3 | 1.6 | -0.6 | 1.5 | 0.7 | 0.1 | 1.5 | 3.0 | -1.3 |
|  | 19:25 | 0.4 | 1.8 | -2.4 | 0.8 | 2.1 | -5.8 | 0.5 | -0.8 | -5.0 | 0.8 | 1.1 | -2.7 | -0.5 | 0.1 | -2.3 | 0.1 | 1.1 | -0.9 | -0.4 | 3.0 | -2.2 |
| 09 July 2020 | 6:55 | 4.0 | 1.3 | -0.8 | 3.3 | 2.2 | -3.3 | 3.7 | 0.5 | -2.6 | 4.4 | 1.6 | -1.1 | 3.5 | 1.6 | 0.1 | 4.4 | 1.6 | 2.5 | 4.1 | 2.7 | -0.6 |
| 10 July 2020 | 6:58 | 4.0 | 1.5 | -0.3 | 3.3 | 2.6 | -3.4 | 3.8 | 0.3 | -2.8 | 4.2 | 1.4 | -0.3 | 2.9 | 1.2 | 0.4 | 4.2 | 0.2 | 0.9 | 4.1 | 2.9 | -0.7 |
| 01 September 2020 | 7:56 | 3.0 | 0.0 | -1.0 | 2.0 | 3.0 | -7.0 | 2.0 | 0.0 | -6.0 | 3.0 | 3.0 | -1.0 | 2.0 | -1.0 | 0.0 | 3.0 | 3.0 | 1.0 | 3.0 | 3.0 | -1.0 |
| 05 September 2020 | 6:45 | 4.0 | 1.0 | -2.0 | 3.0 | 3.0 | -8.0 | 4.0 | 1.0 | -7.0 | 5.0 | 2.0 | -2.0 | 4.0 | 0.0 | -1.0 | 5.0 | 2.0 | 0.0 | 4.0 | 3.0 | -1.0 |
| 10 September 2020 | 7:12 | 5.0 | 1.0 | -2.0 | 3.0 | 2.0 | -8.0 | 4.0 | 0.0 | -8.0 | 5.0 | 2.0 | -2.0 | 4.0 | 0.0 | -1.0 | 5.0 | 3.0 | 0.0 | 4.0 | 2.0 | -2.0 |
| 16 September 2020 | 7:19 | 4.0 | 1.0 | -1.0 | 3.0 | 3.0 | -8.0 | 3.0 | 2.0 | -7.0 | 4.0 | 3.0 | -2.0 | 3.0 | 0.0 | 0.0 | 5.0 | 4.0 | -1.0 | 4.0 | 3.0 | -1.0 |
| 20 September 2020 | 7:50 | 4.0 | 1.0 | -3.0 | 3.0 | 2.0 | -9.0 | 3.0 | -1.0 | -9.0 | 5.0 | 1.0 | -4.0 | 3.0 | 0.0 | -2.0 | 4.0 | 1.0 | -1.0 | 4.0 | 2.0 | -2.0 |
| 25 September 2020 | 7:53 | 4.0 | 1.0 | -3.0 | 2.0 | 2.0 | -9.0 | 3.0 | 0.0 | -10.0 | 4.0 | 1.0 | -3.0 | 3.0 | 0.0 | -2.0 | 4.0 | 2.0 | -2.0 | 3.0 | 3.0 | -2.0 |
| 30 September 2020 | 8:00 | 6.0 | 1.0 | -3.0 | 4.0 | 2.0 | -11.0 | 5.0 | 0.0 | -10.0 | 6.0 | 1.0 | -4.0 | 5.0 | 0.0 | -3.0 | 6.0 | 1.0 | -2.0 | 6.0 | 3.0 | -2.0 |
| 05 October 2020 | 7:58 | 7.0 | 1.0 | -3.0 | 5.0 | 2.0 | -9.0 | 6.0 | 1.0 | -10.0 | 7.0 | 2.0 | -3.0 | 7.0 | 0.0 | -1.0 | 7.0 | 2.0 | -1.0 | 6.0 | 3.0 | -2.0 |
| 10 October 2020 | 8:08 | 8.0 | 1.0 | -2.0 | 5.0 | 2.0 | -9.0 | 7.0 | 0.0 | -9.0 | 7.0 | 1.0 | -3.0 | 7.0 | 0.0 | -2.0 | 7.0 | 3.0 | -1.0 | 7.0 | 3.0 | -2.0 |
| 15 October 2020 | 7:40 | 8.0 | 1.0 | -1.0 | 6.0 | 2.0 | -8.0 | 7.0 | 0.0 | -10.0 | 7.0 | 1.0 | -2.0 | 8.0 | 0.0 | -1.0 | 7.0 | 1.0 | 0.0 | 8.0 | 3.0 | -1.0 |

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
