# Peer review of "A BIM-Based Method for Structural Stability Assessment and Emergency Repairs of Large-Panel Buildings Damaged by Military Actions and Explosions: Evidence from Ukraine"

_buildings, doi:10.3390/buildings12111817_

Round 1

Reviewer 1 Report

The manuscript’s topic might be of sufficient interest and a hot topic, especially due to the ongoing Russia and Ukraine wars. The manuscript’s structure and organization are overall acceptable but can be improved. Also, there are irrelevant and off-topic contents throughout the manuscript, which may distract readers. The manuscript’s innovation is debatable to some degree and needs to be further elaborated and justified. Moreover, the manuscript lacks comprehensive verification/validation. The following comments explain the manuscript’s drawbacks in detail and help improve its quality if addressed meticulously by the authors:

The authors may need to first offer a comprehensive definition for large-panel buildings to clarify it for readers.

References used throughout the entire manuscript are either dated or not scientific (not from journal articles or conference papers with a DUI). The authors may need to add a couple of most-recent scientific research studies relevant to the manuscript’s topic to enhance its quality, especially from a scientific perspective.

This reviewer can neither confirm nor reject the authors’ claims regarding the novelty of this research. The authors claimed that most previous studies focused on developing methods for surveying buildings damaged by natural disasters (earthquakes, hurricanes) and non-military attacks and developed their research based on this claim. The authors may need to provide further justifications for their claims. Given that the authors almost cited dated publications, it is necessary for them to prove the innovation of this research and its main contribution to the body of knowledge.

Parts of the manuscript’s content seem off-topic and unnecessary. For instance, the distribution of the housing stock in Ukraine seems irrelevant to the manuscript’s topic, which is the application of BIM in structural stability assessment and emergency repairs of buildings.

The manuscript lacks a comprehensive validation/verification in which the developed approach for structural stability assessment and emergency repairs is compared with the existing approaches for the same purpose. The authors need to take this into account seriously to improve the quality of their research.

Author Response

The author’s team thanks you for the thorough review of the manuscript. Based on the comments received, some sections of the paper were rewritten, the number of references was increased by more than 20, and the structure was modified. We believe that this both improved the quality of the paper and increased its readability. Please find below the point-by-point responses to your comments

Reviewer 2 Report

From my point of view, this text presents what could be the foundations of the presentation of an interesting crucial research project. However, this present text is not acceptable to be published and this is mainly because it lacks clarity in its contents presentation and it lacks a proper definition of the research limits as well as a proper contextualization.

The reviewer understands that, as the authors recognize in several parts such as the discussion, there is a war undergoing and this war limits the possibilities of the research paper. However, the reviewer considers that the authors can find feasible strategies at hand to solve this paper main lacks due to the war context. An example is that the authors used as case study a 2020 gas-explosion affected building instead of a war affected building.

a) Regarding the definition of the research limits the present text includes a too wide range of topics - from post disaster housing to digital twining - and tries to propose new material in many of them lacking rigor in a high number of them. The reviewer advices the authors to further limit the topic of the article, the amount of contributions, save material for other articles and increase the rigor and analysis in the remaining topic. For instance, the authors could focus on “A BIM-based method for structural stability assessment an emergency repairs of large-panel buildings damaged by gas explosion and its future applicability to war damages”.

b) Regarding the contextualization, this text lacks a proper introduction to the chosen topic, explaining the context from general to particular, from international to national level, from former applications at worldwide level to the specific case study the authors choose, explaining the more relevant related previous technical literature. The technical literature at international level should also improve, also related to the specific topic the authors decide to analyze.

c) The general confusion of the contents and even of the manuscript structure is related to the previous issues a) and b). Apart from the following contents, an explicit presentation of this article objectives and novelty within its field of expertise would also guide potential readers. Then obviously the text should focus on these main objectives and novelty. Also related is the fact that in some parts the writing is not direct as expected in scientific articles. For example, section 2 about methods would highly improve if it gave at the beginning a general explanation about the methodology, steps, tools used in the project. In this general explanation, a graphical figure or schema would help readers.

Author Response

(The authors gave the same response as above.)

Reviewer 3 Report

Well, the Russian aggression is wrong, but this should be scientific paper and not history lesson. Therefore, I recommend to avoid use of some exclamation, e.g. the description of Figure 1 should be

The main areas of problems facing the building stock of Ukraine as a result during the war.

Figure 10 – those deformations in the beginning are after explosion before the placement of the stabilization steel grid?

The prefabricated, or as you stated the large panel buildings are highly dependable on the quality of the joints between the panels – if the joints are properly made: welding of the steel and then the concrete/cement mortar to avoid the corrosion. I did not find mention of this and if/how this is incorporated in the evaluation.

Anyway, the idea of the preparation of different typified systems (completely in 3D) is great and easy to remove some damaged parts – but I consider this mostly as an structural engineering problem, not such much scientific problem.  

Author Response

(The authors gave the same response as above.)

Reviewer 4 Report

Manuscript ID: buildings-1959818

Comments to the Authors

I have now completed my review of the manuscript titled "A BIM-based method for structural stability assessment an emergency repairs of large-panel buildings damaged by military action" The article describes the findings of the analysis and the implications for using 12 building information modeling (BIM) in selecting effective organizational, technological, and technical solutions for eliminating emergency destruction of large-panel buildings as a result of military actions.

Overall, the work has the potential to make a valuable contribution to the journal and will be of interest to its intended audience; nevertheless, there is a need to enhance the research contextually and conceptually on several critical issues. These issues must be addressed and clarified to improve the paper's quality further. The comments on the individual issues are offered below for the author's consideration.

1.      The research question of the study is somewhat missing. I suggest the theoretical positioning is strengthened at the paper's outset, stating why this study is critical to the literature and providing the study's research question. How was the research question answered?

2.      What is the research gap in this study? The authors should add to the manuscript.

3.      The authors should review more detailed theoretical literature on military actions affecting building damage.

4.      The authors should add more detail to the theoretical literature on the BIM-based method for structural stability assessment and emergency repairs.

5.      The authors should add more elaborated content on the digital twin of these case studies in the manuscript.

6.      The authors should explain more in detail the collapse mechanism of the actual damage occurring from the blast.

7.      The authors should explain in detail the BIM Industry Foundation Classes (IFC) of the case studies in the manuscript.

8.      In conclusion, the authors need to elaborate more on structural stability assessment and emergency repairs of large-panel buildings. The authors should describe the practical results of the BIM-based information-mathematical model.

9.      What are the future directions of this study? The authors should address this issue in the manuscript.

10. Practical implications should explain how the BIM models could be used to address building rehabilitation in post-war times.

The reviewer hopes that the comments mentioned above can assist the authors in improving their paper.

Author Response

(The authors gave the same response as above.)

Round 2

Reviewer 1 Report

The comments have been addressed to an acceptable degree.

Author Response

The authors' team thanks you for your review and help in improving the article.

Reviewer 2 Report

From my point of view, this text presents what could be the foundations of the presentation of an interesting crucial research project. However, this present text is not acceptable to be published and this is mainly because it lacks clarity in its contents presentation and it lacks a proper definition of the research limits as well as a proper contextualization.

The reviewer understands that, as the authors recognize in several parts such as the discussion, there is a war undergoing and this war limits the possibilities of the research paper. However, the reviewer considers that the authors can find feasible strategies at hand to solve this paper main lacks due to the war context. An example is that the authors used as case study a 2020 gas-explosion affected building instead of a war affected building.

a) Regarding the definition of the research limits the present text includes a too wide range of topics - large panel buildings damaged by military action, to gas explosions, to post disaster housing - and tries to propose new material in many of them lacking rigor in a high number of them. The reviewer advices the authors to further limit the topic of the article, the amount of contributions, save material for other articles and increase the rigor and analysis in the remaining topic. For instance, the authors could focus on “A BIM-based method for structural stability assessment an emergency repairs of large-panel buildings damaged by gas explosion and its future applicability to war damages”. In any case the title requires rewriting because is giving false expectations. Otherwise, the manuscript and the research project must change drastically to fulfill the current title expectations.

b) Regarding the contextualization, this text lacks a proper introduction to the chosen topic, explaining the context from general to particular, from international to national level, from former applications at worldwide level to the specific case study the authors choose. It is very important to be able to make the text understandable to follow this order, from the general global international overview to the specific case study. The explanation about the more relevant related previous technical literature at international has improved. Nevertheless, it is not a literature review, it does not follow any literature reviews methodology and, therefore, the authors should present it as a general analysis of the main previous technical literature related to the topic.

c) The general confusion of the contents and even of the manuscript structure is related to the previous issues a) and b). Apart from the following contents, an explicit presentation of this article objectives and novelty within its field of expertise would also guide potential readers. Then obviously the text should focus on these main objectives and novelty.

Author Response

(The authors gave the same response as above.)

Reviewer 3 Report

Thank you for the revised version. Sometimes it is better to provide the doc file, to be able to show/hide the track changes.

The article is improved, but the logical process of the paper can be improved.

In the figure 11, why there are no dots for the measured values for the july for example?just the values are connected together from 30.6 to 20.9.

How do you reflect the differences after reinforcement of the building? Some are still negative, but other are positive.

The mention of the joints is not good enough in my oppinion, should be also mentioned in the inroduction and description of the panel building stock. How was the the joints considered in the modeling of the Fig. 9 building?

Author Response

(The authors gave the same response as above.)

Reviewer 4 Report

The comments have been addressed to an acceptable level. The improved paper is well-written and structured.

Author Response

(The authors gave the same response as above.)

Round 3

Reviewer 2 Report

From my point of view, this text presents what could be the foundations of the presentation of an interesting crucial research project. However, this present text is not acceptable to be published and this is mainly because it lacks clarity in its contents presentation and it lacks a proper definition of the research limits as well as a proper contextualization.

a) Regarding the definition of the research limits the present text includes a too wide range of topics - large panel buildings damaged by military action, to gas explosions, to post disaster housing - and tries to propose new material in many of them lacking rigor in a high number of them. For example, Figure 1 gives information about temporary housing and post-war reconstruction that lacks further definition of terms, concepts, acronyms, lacks relying on technical literature, etc, to be scientifically rigorous enough. The reviewer advices the authors to further limit the topic of the article, the amount of contributions, save material for other articles and increase the rigor and analysis in the remaining topic. For instance, the authors could focus on “A BIM-based method for structural stability assessment and emergency repairs of large-panel buildings damaged by gas explosion and its future applicability to war damages”. In any case the title requires rewriting because is giving false expectations. Otherwise, the manuscript and the research project must change drastically to fulfill the current title expectations. The present title introduces the explosions, which is the type of damage developed in depth in this paper. So, in this sense the article has improved although it still lacks giving real account of what the article is about.

b) Regarding the contextualization, this text lacks a proper introduction to the chosen topic, explaining the context from general to particular, from international to national level, from former applications at worldwide level to the specific case study the authors choose. It is very important to be able to make the text understandable to follow this order, from the general global international overview to the specific case study. The reviewer thinks the authors are not understanding this important lack. This lack is about providing a general introduction to the topic before starting with the local context and temporal context in the very first line of the introduction (line 33). Ukraine military actions from 2022 are the case study. The authors lack contextualizing this case study within the global and international context. Are there other scenarios that could benefit from this study of structural stability assessment and emergency repairs of large-panel buildings damaged by gas explosions?

c) The general confusion of the contents and even of the manuscript structure is related to the previous issues a) and b). Apart from the following contents, an explicit presentation of this article objectives and novelty within its field of expertise would also guide potential readers. Then obviously the text should focus on these main objectives and novelty. This comment refers to the introduction, and specifically of its last paragraph in lines 227-233, which lacks clarity in explaining the research objectives and research gap or contribution. From the reviewer point of view this paragraph lacks clarity and coherence with the research project and the manuscript contents.

Author Response

Dear reviewer,

Our point-by-point responses to your comments are below:

Point 1: a) Regarding the definition of the research limits the present text includes a too wide range of topics - large panel buildings damaged by military action, to gas explosions, to post disaster housing - and tries to propose new material in many of them lacking rigor in a high number of them. For example, Figure 1 gives information about temporary housing and post-war reconstruction that lacks further definition of terms, concepts, acronyms, lacks relying on technical literature, etc, to be scientifically rigorous enough. The reviewer advices the authors to further limit the topic of the article, the amount of contributions, save material for other articles and increase the rigor and analysis in the remaining topic. For instance, the authors could focus on “A BIM-based method for structural stability assessment and emergency repairs of large-panel buildings damaged by gas explosion and its future applicability to war damages”. In any case the title requires rewriting because is giving false expectations. Otherwise, the manuscript and the research project must change drastically to fulfill the current title expectations. The present title introduces the explosions, which is the type of damage developed in depth in this paper. So, in this sense the article has improved although it still lacks giving real account of what the article is about.

Response 1: We edited the title of the article noting that only the experience of Ukraine is considered. The authors’ team decided that there could be two approaches to address the issue of the wide range of the problem – either by narrowing down the research question, or indicating that a complex problem is discussed in this paper, but focused on Ukraine and the ongoing war. We chose the latter. This way, we believe, the indications in what are the current problems of the construction industry in Ukraine and what exactly we are investigating in this paper, become much more clear for the reader.

Thank you for noting the abbreviations in Figure 1, the acronyms in the schematics were replaced by full titles.

Point 2: b) Regarding the contextualization, this text lacks a proper introduction to the chosen topic, explaining the context from general to particular, from international to national level, from former applications at worldwide level to the specific case study the authors choose. It is very important to be able to make the text understandable to follow this order, from the general global international overview to the specific case study. The reviewer thinks the authors are not understanding this important lack. This lack is about providing a general introduction to the topic before starting with the local context and temporal context in the very first line of the introduction (line 33). Ukraine military actions from 2022 are the case study. The authors lack contextualizing this case study within the global and international context. Are there other scenarios that could benefit from this study of structural stability assessment and emergency repairs of large-panel buildings damaged by gas explosions?

Response 2: Thank you for clarifying your comment. We made the following changes to the manuscript – 1) in the title of the paper we used the addition “evidence from Ukraine”, and 2) a contextualization chapter was added to the introduction section (lines 33-46). This helped to have a better alignment for the readers between the title, introduction, and main text.

The main goal of this study is placing foundations towards the development of a methodology for assessing the damaged apartment buildings, which have been the most affected. Considering that the main part of the damaged buildings is located close to the combat zone and there is currently no access to them, the main survey work is still ahead. Therefore, it is important to develop methods for their further use now.

When writing the introduction and background for this study, we found that there’s a significant lack of information in this direction since cases of such massive damage to residential buildings were rare and publications in this direction are very few. The modified introduction and the title of the manuscript, hopefully, will help the reader to better understand, that we’re looking at a global problem, but the solutions should be local.

 Point 3: c) The general confusion of the contents and even of the manuscript structure is related to the previous issues a) and b). Apart from the following contents, an explicit presentation of this article objectives and novelty within its field of expertise would also guide potential readers. Then obviously the text should focus on these main objectives and novelty. This comment refers to the introduction, and specifically of its last paragraph in lines 227-233, which lacks clarity in explaining the research objectives and research gap or contribution. From the reviewer point of view this paragraph lacks clarity and coherence with the research project and the manuscript contents.

Response 3: Thank you for explaining the question. We have changed this paragraph. We hope that our research objectives and contribution have become clearer.

Kind regards,

Authors' team

Reviewer 3 Report

thank you for noticing my comments.

The question about the change of the deformation after reinforcement was more about what or why some points still moves to minus and some were e.g. towards plus. 

Author Response

Dear reviewer,

Our response to your comment is below:

Point 1: The question about the change of the deformation after reinforcement was more about what or why some points still moves to minus and some were e.g. towards plus.

Response 1: Thank you for clarifying the question. The main goal of geodetics observation was to monitor only the dynamics of the deformation processes of the damaged building. That’s why we did not perform a detailed analysis of the change in the position of the deformation marks for this case study.

But in general, for this damaged building, the deformations were not uniform, the marks changed their positions at different speeds and in different directions. Indeed, the displacement of the deformation marks along the X and Y axes took on positive values ​​(orientation of the axes in Figure 9a) after the strengthening. As a result of damage to the integrity of the building frame, initially part of the panels began to move towards the epicenter of the damage. At the same time, part of the panels was shifted to the outside of the building along the Y-axis. After the installation of the metal frame, they returned closer to their previous design position. The stable dynamics of positive values ​​along the X-axis may indicate the development of the building's tilt in this direction, but it was not investigated separately at that time. The most critical is the displacement of deformation marks along the vertical axis H - subsidence. The values ​​of subsidence of deformation marks were consistently negative, although the dynamics of their change were different. That’s indicating that the reinforcement inhibited the process of vertical movement of the panels and stabilized the buildings for further dismantling.

At the same time, we emphasize that the main task was to monitor the dynamics of the development of deformations in order to prevent the uncontrolled collapse of structures. Therefore, the analysis of the absolute values ​​of deformation marks and their behavior before and after strengthening was not performed. For such a task, work methods and equipment would have to be different, which would lead to an increase in the cost of work. Nevertheless, the data we have collected is insufficient for such an analysis to describe the behavior of marks, therefore, we do not present this analysis in our article.

Kind regards,

Authors' team